# Realization of nearly dispersionless bands with strong orbital anisotropy from destructive interference in twisted bilayer MoS$_2$

Lede Xian [1,2], Martin Claassen[3,4], Dominik Kiese[5], Michael M. Scherer[5], Simon Trebst [5], Dante M. Kennes [1,6 ✉] & Angel Rubio [1,3,7 ✉]

Recently, the twist angle between adjacent sheets of stacked van der Waals materials emerged as a new knob to engineer correlated states of matter in two-dimensional hetero-structures in a controlled manner, giving rise to emergent phenomena such as super-conductivity or correlated insulating states. Here, we use an ab initio based approach to characterize the electronic properties of twisted bilayer MoS$_2$. We report that, in marked contrast to twisted bilayer graphene, slightly hole-doped MoS$_2$ realizes a strongly asymmetric p$_x$-p$_y$ Hubbard model on the honeycomb lattice, with two almost entirely dispersionless bands emerging due to destructive interference. The origin of these dispersionless bands, is similar to that of the flat bands in the prototypical Lieb or Kagome lattices and co-exists with the general band flattening at small twist angle due to the moiré interference. We study the collective behavior of twisted bilayer MoS$_2$ in the presence of interactions, and characterize an array of different magnetic and orbitally-ordered correlated phases, which may be sus-ceptible to quantum fluctuations giving rise to exotic, purely quantum, states of matter.

---

[1] Max Planck Institute for the Structure and Dynamics of Matter and Center for Free Electron Laser Science, Hamburg, Germany. [2] Frontier Research Center, Songshan Lake Materials Laboratory, Dongguan, China. [3] Center for Computational Quantum Physics, Simons Foundation Flatiron Institute, New York, NY, USA. [4] Department of Physics and Astronomy, University of Pennsylvania, Philadelphia, PA, USA. [5] Institute for Theoretical Physics, University of Cologne, Cologne, Germany. [6] Institut für Theorie der Statistischen Physik, RWTH Aachen University and JARA-Fundamentals of Information Technology, Aachen, Germany. [7] Nano-Bio Spectroscopy Group, Departamento de Fisica de Materiales, Universidad del País Vasco, San Sebastián, Spain. ✉email: dante.kennes@rwth-aachen.de; angel.rubio@mpsd.mpg.de

Two-dimensional van der Waals materials constitute a versatile platform to realize quantum states by design, as they can be synthesized in many different stacking conditions[1], offer a wide variety of chemical compositions, and are easily manipulated by back gates, strain and the like. Stacking two sheets of van der Waals materials atop each other at a relative twist has recently emerged as a vibrant research direction to enhance the role of electronic interactions, with first reports on twisted bilayer graphene[2–6] and another van der Waals materials stacked atop each other at a twist[7–17] displaying features of correlated physics that afford a high level of control. In particular, bi-, tri-, and quadruple-layer graphene[18] as well as twisted few-layer transition metal dichalcogenides (TMDs)[19,20] are currently under intense experimental scrutiny[13,21–29]. By forming a moiré supercell at small twist angles, a large unit cell in real space emerges for twisted systems, which due to quantum interference effects leads to a quasi-two-dimensional system with strongly quenched kinetic energy scales. This reduction in kinetic energy scale, signaled by the emergence of flat electron bands, in turn enhances the role of electronic interactions in these systems. Therefore, twisted systems enable the realization of new correlated condensed matter models, establishing a solid-state quantum simulator platform[30].

Whereas the flatting of band dispersions in two-dimensional moiré superlattices results mainly from the localization of charge density distributions by the moiré potential, a well-known alternate pathway to flat bands can occur in certain lattices such as the Lieb and the Kagome lattices. Here, purely geometric considerations lead to the formation of perfectly localized electronic states that have weight only on single plaquettes or hexagons, respectively, and that are eigenstates of the kinetic Hamiltonian due to destructive interference between lattice hopping matrix elements[31]. To put it differently, linear combinations of the macroscopically degenerate extended Bloch states in these systems allows to form localized Wannier-like eigenstates (living on single plaquettes or hexagons in the examples above) with no dispersion (for a review on the subject see, e.g,[32]). Such flat band systems can give rise to many interesting phenomena, such as the formation of nontrivial topology when time-reversal symmetry is broken, or other exotic quantum phases of matter due to their susceptibility to quantum fluctuations and electronic correlations[32].

Here, we demonstrate that both flat band mechanisms can be engineered to coexist in twisted bilayers of $MoS_2$ (tb$MoS_2$): a TMD of direct experimental relevance that has been extensively studied from synthesis to applications[33,34]. We confirm that families of flat bands emerge when two sheets of $MoS_2$ in the 2H structure are stacked at a twist[12,35] due to moiré potentials. Our large-scale ab initio based simulations show that while the first set of engineered flat bands closest to the edge of the bandgap with twist angles close to $\Theta \approx 0°$ can be used to effectively engineer a non-degenerate electronic flat band in analogy to a single layer of graphene at meV energy scales, more intriguingly, the next set of flat bands instead realizes a strongly asymmetric flat band $p_x$–$p_y$ honeycomb lattice[36,37]. Both of these families of bands should be accessible experimentally via gating. The strongly asymmetric nature of this $p_x$–$p_y$ honeycomb lattice is in marked contrast to the much-discussed case of twisted bilayer graphene, where an approximately symmetric version of such a Hamiltonian is now believed to describe the low-energy flat band structures found at small twist angle[38–42]. The strongly asymmetric $p_x$–$p_y$ honeycomb model itself features two almost entirely dispersionless flat bands that touch the top and the bottom of graphene-like Dirac bands at the Gamma point, respectively. These flat bands in this model originate from destructive interference, in analogy to flat bands in the Lieb and the Kagome lattices[31] discussed above, and will be referred to as ultra-flat bands in the following discussion.

On top of that, the total bandwidth of the strongly asymmetric $p_x$–$p_y$ honeycomb effective model realized here (all four bands) can be further flattened by decreasing the twist angle. In addition, these ultra-flat bands can be topologically nontrivial in the presence of spin-orbital coupling (SOC)[43]. Although all the flat bands discussed here originate from the Γ-point states of $MoS_2$ and are not affected by intrinsic SOC (see Supplementary Fig. 3), we expect that substrate engineering[44] can be used to introduce SOC coupling into these bands and invoke topologically nontrivial behavior of the ultra-flat band states. Previously, the $p_x$-$p_y$ model was studied in the context of cold gases where exotic correlated phases were predicted[36,45,46], as well as in semiconductor microcavities[47] and certain 2D systems such as organometallic frameworks[48,49] and Bismuth deposited on SiC[50] with a focus on their nontrivial topology properties. Our findings elevate tb$MoS_2$ to an interesting platform where effects of ultra-flat bands can be studied systematically in a strongly correlated solid-state setting.

Notably, in the strong-coupling regime, the $p_x$–$p_y$ model amended by Hubbard and Hund's interactions gives rise to a spin-orbital honeycomb model which – depending on the specific parameters and symmetries of the model – hosts magnetic, orbital as well as valence-bond orderings, or even more exotic quantum spin-orbital liquid phases[51–53]. With this, our work adds an interesting type of lattice model – the highly asymmetric $p_x$–$p_y$ Hubbard model – to the growing list of systems that can effectively be engineered using the twist angle between multiple layers. This is particularly intriguing as we maintain the full advantages that come with two-dimensional van der Waals materials, such as relative simplicity of the chemical composition and controllability of the material properties; e.g. of the filling (by a back gate), electric tunability (by displacement fields) or the bandwidth of the model (by the twist angle).

## Results

**Ab initio characterization of twisted $MoS_2$.** We first characterize the low-energy electronic properties of twisted bilayer $MoS_2$ using density functional theory (DFT) calculations (see Methods). DFT in particular has established itself as a reliable tool to provide theoretical guidance and to predict the band structure of many twisted bi- and multilayer materials[8,13,15]. However, such a first-principles characterization becomes numerically very demanding as the twist angle Θ approaches small values and the unit cell becomes very large entailing many atoms (of the order of a few thousands and more). Nevertheless, it is that limit in which strong band-narrowing effects and as a consequence prominent effects of correlations are expected. The results of such a characterization are summarized in Fig. 1. Note that atomic relaxation has been shown to affect the electronic properties of twisted 2D materials[12,35,54]. While for twisted bilayer graphene this effect is only significant at twist angles smaller than 1 degree[54], it noticeably alters the low-energy band dispersions and charges density localization for twisted transition metal dichalcogenides bilayer (such as $MoS_2$) even with relatively large twist angles above 1 degree[12,35]. Therefore, we relax all the systems in our DFT calculations. Panel (a) shows the relaxed atomic structure of two sheets of $MoS_2$ in real space, twisted with respect to each other. A moiré interference pattern forms at a small twist angle yielding a large unit cell, within which we identify different local patterns of stacking of the two sheets of $MoS_2$, indicated via areas framed by cyan, magenta or purple dashed lines. The local stacking arrangements of the respective areas are given in the right sub-panels of the panel (a). Note that the $B^{\bar{M}o/S}$ and the $B^{S/Mo}$ regions are equivalent as they are related by C2 symmetry. These equivalent $B^{\bar{M}o/S}/B^{Mo/S}$ regions form a hexagonal network.

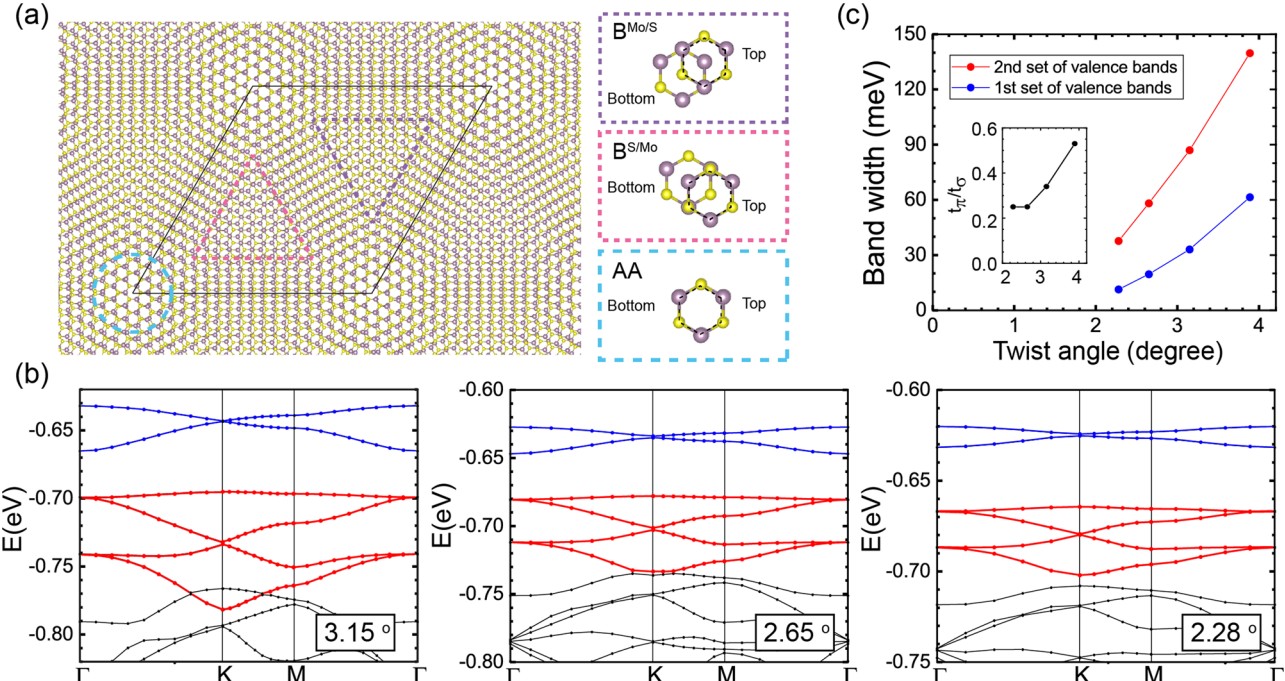

**Fig. 1 Atomic and electronic structures of twisted bilayer MoS$_2$. a** Atomic structure of tbMoS$_2$ at $\Theta = 3.15°$. Local atomic arrangements of the three different regions in the moiré unit cell are indicated in the right panels. The Mo (S) atoms are indicated with purple (yellow) balls. **b** Evolution of low-energy band structures at the top of the valence bands of tbMoS$_2$ with decreasing small twist angles. The first set and the second set of valence bands are highlighted with blue and red lines, respectively. **c** Evolution of the bandwidth of the first set and the second set of valence bands with decreasing twist angles. Inset: twist angle dependence of the ratio of the hopping amplitudes $t_\pi$ and $t_\sigma$ in the p$_x$–p$_y$ honeycomb lattice.

In panel (b) we show the ab initio band structure of the twisted material after relaxation, where we find two families of bands that will become increasingly flat and start to detach from all other bands, as the twist angle is lowered. We mark these bands by blue and red color in panel (b), which shows results for decreasing angles from $\Theta = 3.16°$–$2.28°$. The bandwidth of these two energetically separated groups of bands is summarized in panel (c) of Fig. 1. We find that the bandwidth of these two bands shrinks drastically as the angle is decreased, yielding bandwidths of the order of 10 meV as the angle approaches $\Theta \approx 2°$. Similar features are also shown in the work of Naik et al.[35]. The bandwidth and the shape of the flat bands (in particular for the second set) in our calculations are slightly quantitatively different from the previous work probably because we relax the structure directly with DFT while the authors of ref. [35] use a force-field approach. Note that these flat bands near the top of the valence bands originate from the states around the $\Gamma$ point in the Brillouin zone of the primitive unit cell of untwisted MoS$_2$, with both S p$_z$ and Mo d$_{z^2}$ characters (see Supplementary Fig. 2 for a DFT characterization of the orbital contribution to the different bands). This is different to the case of twisted WSe$_2$, where the top valence flat bands originate from the states around the K point in the Brillouin zone of the primitive unit cell (dominated by W d$_{x^2-y^2}$ and d$_{xy}$ orbitals), which experience different interlayer moiré potentials compared with those of the $\Gamma$-point flat bands discussed here leading to an effective triangular lattice Hubbard model[13]. Since also in other TMDs, such as MoSe$_2$ and WS$_2$, the top of the valence band in the untwisted bilayer is also located at the $\Gamma$ point in the Brillouin zone[55,56], the physics we discussed here transfers also to those materials being twisted.

The upper bands in Fig. 1 (marked in blue) show a Dirac cone at the K point and behave very similar to the bands found for monolayer graphene (with the exception of a reduced

bandwidth). They are spin degenerate in nature, but feature no additional degeneracy except at certain high symmetry points. Instead, the next set of bands (marked in red) is essential to our work. They too feature a Dirac cone at the K point, but also feature two additional ultra-flat bands at the top and bottom in addition to a band structure similar to graphene. The ratio between the width of the ultra-flat and the flat bands decreases as the angle is decreased, but saturates in our calculations as a twist angle of $\Theta \approx 2.28°$ is approached. We attribute this saturation to lattice relaxation effects; note however that the overall bandwidth keeps decreasing. To access this second set of bands we need to empty the bands marked in blue first. The effects of this doping are of minor quantitative nature (see Supplementary Fig. 5).

Remarkably, this second family of flat bands is well-described by an effective p$_x$–p$_y$ tight-binding model on a honeycomb lattice, depicted schematically in Fig. 2a, and conveniently described by the following Hamiltonian:

$$H_0 = \sum_{\langle i,j \rangle,s} (t_\sigma \mathbf{c}_{i,s}^\dagger \cdot \mathbf{n}_{ij}^\parallel \mathbf{n}_{ij}^\parallel \cdot \mathbf{c}_{j,s} - t_\pi \mathbf{c}_{i,s}^\dagger \cdot \mathbf{n}_{ij}^\perp \mathbf{n}_{ij}^\perp \cdot \mathbf{c}_{j,s})$$
$$+ \sum_{\langle\langle i,j \rangle\rangle,s} (t_\sigma^N \mathbf{c}_{i,s}^\dagger \cdot \mathbf{n}_{ij}^\parallel \mathbf{n}_{ij}^\parallel \cdot \mathbf{c}_{j,s} - t_\pi^N \mathbf{c}_{i,s}^\dagger \cdot \mathbf{n}_{ij}^\perp \mathbf{n}_{ij}^\perp \cdot \mathbf{c}_{j,s}), \quad (1)$$

where $\mathbf{c}_{i,s} = (c_{i,x,s}, c_{i,y,s})^T$ with $c_{i,x(y),s}$ annihilating an electron with p$_{x(y)}$-orbital at site $i$ and with spin $s = \uparrow, \downarrow$. $\langle i,j \rangle$ ($\langle\langle i,j \rangle\rangle$) denotes (next) nearest neighbors. For each sum in Eq. (1), the first term describes the $\sigma$ hopping (head to tail) between the p-orbitals and the second term denotes the $\pi$ hopping (shoulder to shoulder). Furthermore, $\mathbf{n}_{ij}^\parallel = (\mathbf{r}_i - \mathbf{r}_j)/|\mathbf{r}_i - \mathbf{r}_j|$, with $\mathbf{r}_i$ being the position of site $i$ and $\mathbf{n}_{ij}^\perp = U\mathbf{n}_{ij}^\parallel$ with $U$ being the two-dimensional 90 degree rotation matrix $U = \begin{pmatrix} 0 & -1 \\ 1 & 0 \end{pmatrix}$. Finally, $t_\sigma$ and $t_\pi$ ($t_\sigma^N$ and $t_\pi^N$) are the nearest neighbor (next-nearest neighbor) hopping amplitudes

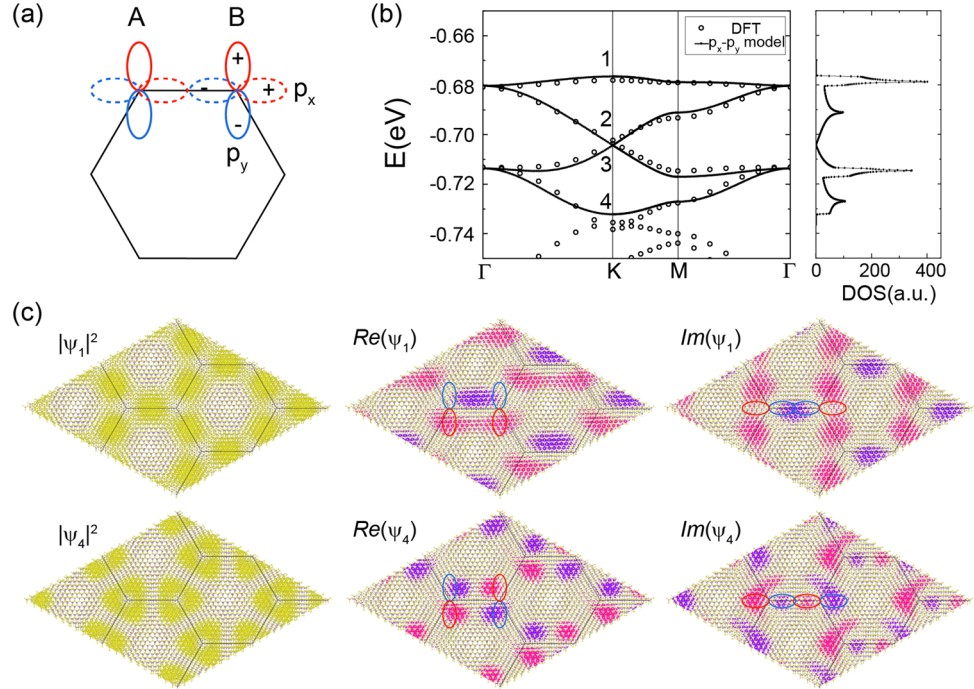

**Fig. 2 $p_x$-$p_y$ honeycomb model for twisted bilayer MoS₂. a** Illustration of the model: in a honeycomb lattice composed of sublattices A and B, there are two orthogonal orbitals ($p_x$ and $p_y$) at each of the two sublattice sites. The solid and the dashed lines denote the $p_y$ and the $p_x$ orbitals, respectively, and the red and the blue color denotes the positive and the negative side of the orbital, respectively. **b** Fitting the dispersion of the $p_x$-$p_y$ model to the second set of valence bands of tbMoS₂ calculated with DFT for tbMoS₂ at 2.65°. The left panel shows the corresponding density of states displaying the signature four-peak structure. **c** Charge density, real and imaginary parts of the wave function calculated with DFT for the states in the two quasi-flat bands 1 and 4 shown in (**b**). The isosurface of the charge density is colored yellow. The positive and the negative parts of the isosurfaces of the wave function are colored in pink and purple, respectively. The solutions of the corresponding states from the $p_x$–$p_y$ model are indicated with the blue and red ovals and agree with the DFT results.

for the $\sigma$-bonding term and $\pi$-bonding term, respectively. Figure 2b, c depict the corresponding dispersions, density of states, and wave functions in comparison to model predictions, illustrating that the four moiré bands at low energies are well captured by Eq. (1) upon the choice of hopping parameters $t_\pi = 0.25 t_\sigma$, $t_\sigma^N = 0.07 t_\sigma$ and $t_\pi^N = -0.04 t_\sigma$. The density of states exhibits a characteristic four van Hove singularities structure, with two originating from the Dirac bands and two stemming from the additional two ultra-flat bands. The small ratio between the nearest neighbor hopping amplitudes $t_\pi/t_\sigma$ determines the residual small dispersion in the ultra-flat bands we report. This ratio is controllable by the twist angle, which is summarized in the inset of Fig. 1c. All these parameters are related to the interlayer moiré potential and are thus expected to be also affected and controllable by the uniaxial pressure perpendicular to the layers as demonstrated for twisted bilayer graphene[4].

The flat band wavefunctions consist of atomic wavefunctions from the $p_z$ orbital on S atoms and the $d_{z^2}$ orbital on Mo atoms. Modulated by the moiré potential, the weighting of the atomic wavefunctions and their modulus square (i.e., charge density) vary at different atomic sites across the whole supercell, showing distinct patterns for different flat band states at the K point in the supercell Brillouin zone as shown in Panel (c) of Fig. 2. These patterns of the charge density as well as the real and the imaginary part of the total wavefunctions obtained from DFT show features consistent with those of the $p_x$–$p_y$ Hamiltonian of Eq. (1). Note, that we call this the $p_x$-$p_y$ Hamiltonian to connect to established literature on the subject; whereas the actual moiré wave functions are composed of $p_z$ and $d_z$ orbitals, they transform like $p_x$, $p_y$ orbitals according to the irreps of the reduced

symmetry group of the moiré supercell. Interestingly, the charge density distribution of the top ultra-flat band state displays a Kagome lattice structure. We have thus unambiguously established twisted MoS₂ to be a candidate system to realize a $p_x$–$p_y$ model on the honeycomb lattice with strongly asymmetric hoppings $t_\sigma$ and $t_\pi$, giving rise to a new set of ultra-flat bands.

**Correlations and magnetic properties.** We now study the role of electronic interactions. As the highly-anisotropic $p_x$-$p_y$ orbital structure constitutes the essential novelty of twisted bilayer MoS₂, we focus on quarter filling (one electron per sublattice in the Moié unit cell) where orbital fluctuations can be expected to be crucial. This filling fraction is straightforwardly accessible in the experiment via back gating, and we defer a discussion of the half-filled case to Supplementary Note 1. To proceed, we assume purely local electronic interactions, which can be generically parameterized in terms of the Hubbard-Kanamori Hamiltonian:

$$H_U = U \sum_{i,\alpha} n_{i\alpha\uparrow} n_{i\alpha\downarrow} + (U - 2J) \sum_i n_{ix} n_{iy} + J \sum_{i,s,s'} c_{ixs}^\dagger c_{iys'}^\dagger c_{ixs'} c_{iys}$$
$$+ J \sum_{i,\alpha\neq\beta} c_{i\alpha\uparrow}^\dagger c_{i\alpha\downarrow}^\dagger c_{i\beta\downarrow} c_{i\beta\uparrow}$$

$$(2)$$

for two orbitals with rotational symmetry. More realistic modelling should include long-range interactions. However, for our choice of commensurate quarter filling, any longer-ranged component of the Coulomb interaction at strong-coupling will serve merely to renormalize the effective spin-orbital interactions of the resulting Kugel-Khomskii model and we therefore concentrate on purely local interactions for simplicity. Furthermore, our DFT

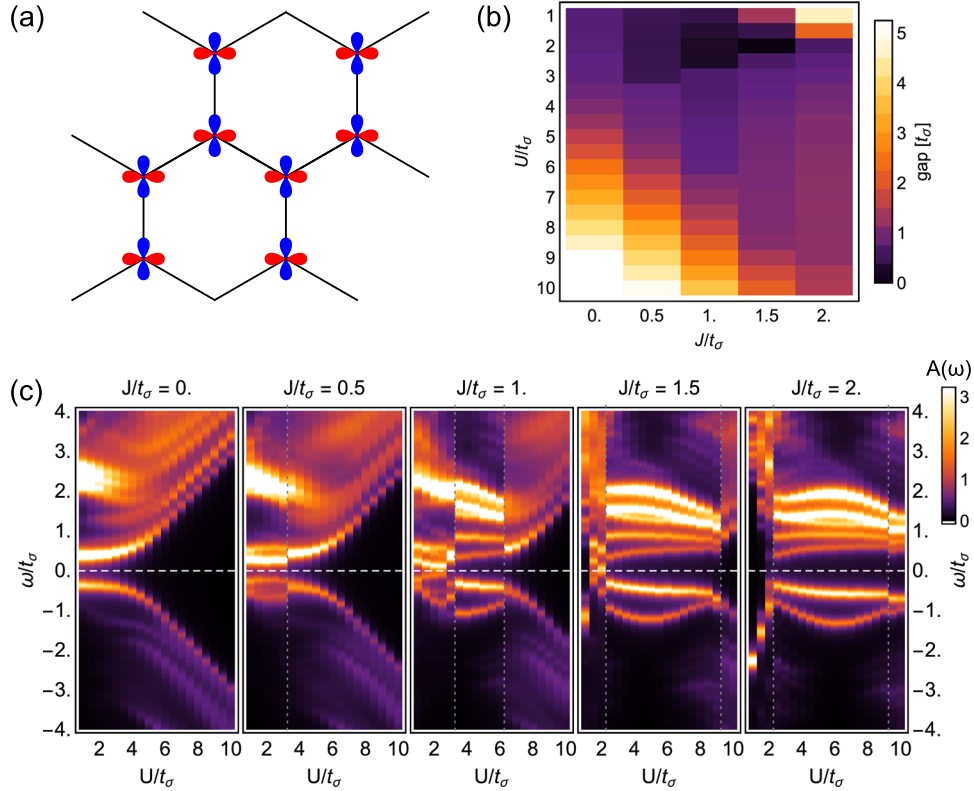

**Fig. 3 Charge gap and correlations for twisted bilayer MoS₂ at vanishing temperature. a** depicts the 16-orbital cluster geometry employed for exact diagonalization of the Hubbard-Kanamori Hamiltonian. **b** depicts the charge gap as a function of Hubbard $U$ and Hund's exchange $J$ interactions, calculated for the 16-orbital cluster and extracted from (**c**) the local density of states, which is readily accessible via scanning tunnelling microscopy. A well-defined charge gap develops beyond $U/t_\sigma \sim 4$ at small $J$ that scales linearly with the Hubbard interaction $U$. Vertical gray dotted lines indicate phase transitions to charge-ordered states at large $J/U$, coinciding with a closing of the charge gap.

calculations suggest $t_\pi \approx 0.25 t_\sigma$ and only weak next-nearest neighbor hopping at small twist angles; we therefore neglect next-nearest neighbor hopping in the analysis below (see Supplementary Fig. 4 for a comparison of the band structures with and without next-nearest neighbor hopping). An ab initio based characterization of the values of $U$ and $J$ requires numerically expansive Wannierzation of the wave functions and is unfortunately beyond the scope of this work. However, by substrate engineering[22] it is likely that a whole range of values can be accessed and therefore it is useful to vary these parameters to explore all possible phases accessible in experiments to make concrete predictions. Vice versa given a future experimental observation our results can be used to estimate the strength of correlations.

Figure 3c depicts the local density of states as a function of Hubbard $U$ and Hund's exchange $J$ interactions, calculated via an exact diagonalization study of Eqs. (1) and (2) for a cluster depicted schematically in (a). Clear evidence of a charge gap beyond $U/t_\sigma \sim 4$ at small $J$ signifies the onset of a correlated insulator which could be directly observed via transport and scanning tunnelling microscopy. The behavior of the gap is depicted in Fig. 3b as a function of $U, J$ and signifies that charge fluctuations are strongly suppressed for large $U$. Establishing the existence of a charge gap motivates to set up a strong-coupling Hamiltonian routinely employed for the types of systems under scrutiny here.

In this regime, a natural follow-up questions concerns possible orderings of the orbital and magnetic degrees of freedom. The corresponding strong-coupling Kugel-Khomskii Hamiltonian[57–59] for the $p_x$-$p_y$ model at quarter filling is given in refs. [51–53] and

reads:

$$H = \sum_{\langle ij \rangle} \frac{1}{U-3J} \xi_{ij}^1 \Big[ t_\sigma t_\pi \bar{Q}_{ij} - (t_\sigma^2 + t_\pi^2)(P_{ij}^{xy} + P_{ij}^{yx}) \Big]$$
$$- \frac{1}{U+J} \xi_{ij}^0 \Big[ t_\sigma t_\pi Q_{ij} + 2t_\sigma^2 P_{ij}^{xx} + 2t_\pi^2 P_{ij}^{yy} \Big]$$
$$+ \frac{1}{U-J} \xi_{ij}^0 \Big[ t_\sigma t_\pi (Q_{ij} - \bar{Q}_{ij}) - 2t_\sigma^2 P_{ij}^{xx} - 2t_\pi^2 P_{ij}^{yy} - (t_\sigma^2 + t_\pi^2)(P_{ij}^{xy} + P_{ij}^{yx}) \Big].$$

(3)

Here, $\xi_{ij}^1 = 3/4 + \mathbf{S}_i \mathbf{S}_j$ denotes the projector onto triplet states, whereas $\xi_{ij}^0 = 1/4 - \mathbf{S}_i \mathbf{S}_j$ selects the singlet spin states instead. Note that the orbital operators, for example $Q_{ij}$, are bond dependent, giving rise to a strong spatial anisotropy of the resulting spin-orbit model. To be more precise following ref. [52], the operators $Q_{ij}$ and $\bar{Q}_{ij}$ describe processes where orbital occupations of sites $i$ and $j$ are reversed, that is they are defined as $Q_{ij} = (\tau_i^+ \tau_j^+ + \tau_i^- \tau_j^-)/2$ and $\bar{Q}_{ij} = (\tau_i^+ \tau_j^- + \tau_i^- \tau_j^+)/2$, with $\tau_i^\pm = \mathbf{n}_{ij}^\perp \boldsymbol{\tau}_i \pm i\tau_i^y$ where $\boldsymbol{\tau}_i = (\tau_i^z, \tau_i^x, \tau_i^y)^T$. The orbital projection operators can then be expressed as $P_{ij}^{xx} = (1 + \mathbf{n}_{ij}^\parallel \boldsymbol{\tau}_i)(1 + \mathbf{n}_{ij}^\parallel \boldsymbol{\tau}_j)/4$, $P_{ij}^{yy} = (1 - \mathbf{n}_{ij}^\parallel \boldsymbol{\tau}_i)(1 - \mathbf{n}_{ij}^\parallel \boldsymbol{\tau}_j)/4$, $P_{ij}^{xy} = (1 + \mathbf{n}_{ij}^\parallel \boldsymbol{\tau}_i)(1 - \mathbf{n}_{ij}^\parallel \boldsymbol{\tau}_j)/4$ and $P_{ij}^{yx} = (1 - \mathbf{n}_{ij}^\parallel \boldsymbol{\tau}_i)(1 + \mathbf{n}_{ij}^\parallel \boldsymbol{\tau}_j)/4$, where e.g. $P_{ij}^{xx}$ selects states where the superposition $(p_x \mathbf{e}_x + p_y \mathbf{e}_y)\mathbf{n}_{ij}^\parallel$ is occupied on nearest neighbor sites $i$ and $j$ connected by the bond $\mathbf{n}_{ij}^\parallel$.

To study its ground state phase diagram using the ab initio parameters found in the previous section, we employ a mean-field analysis of competing for orbital orderings with ferromagnetic and antiferromagnetic spin order. Note, that the simplifying

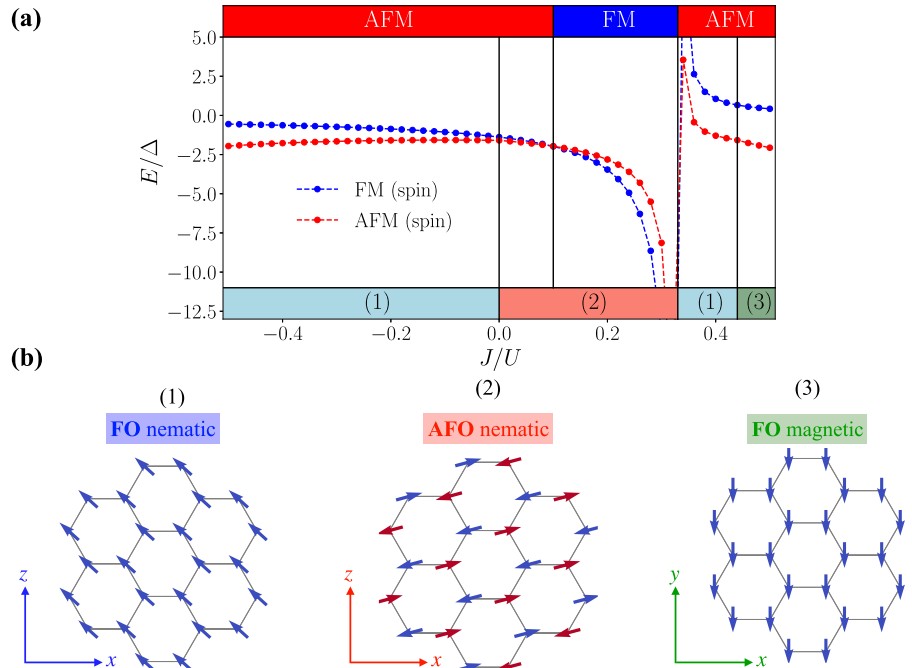

**Fig. 4 Magnetic phase diagram for twisted bilayer MoS₂. a** Classical ground state energy per orbital in units of $\Delta = t_\sigma^2/U$, assuming ferro- (blue) or antiferromagnetic (red) order for the spin degrees of freedom. We take the ab initio parameters, $t_\pi = 0.25 t_\sigma$ and use an iterative energy minimization. The lower panel determines the phase boundaries for the orbital degrees of freedom given the energetically more favorable spin order shown in the top panel. At $J/U = 0.1$ we find the spin order to change from AFM to FM, with AFO nematic order for the orbital degrees of freedom remaining stable in agreement with ref. [53]. **b** Configurations of orbital vectors are found at the end of iterative minimization. Note that we display the projection of $\boldsymbol{\tau}$ to the plane in $\mathbb{R}^3$ (indicated by the axis shown in the bottom left), such that nematic states with finite contributions only in $xz$ direction ((1) & (2)) can be distinguished from magnetic states (3) which point perpendicular, i.e along the $y$-axis.

assumption of vanishing temperature – a standard one in condensed matter research – still allows to draw conclusions for the low-temperature physics accessible in experiments as fingerprints of the phases we discuss extend into this regime as well. To this end, we note that on the bipartite honeycomb lattice the SU(2) invariant spin sector would, on its own, order either ferro- or antiferromagnetically, depending on the sign of the exchange couplings. As an Ansatz, we therefore assume that one of the respective states is stabilized and decouple the spin from the orbital degrees of freedom by replacing $\mathbf{S}_i\mathbf{S}_j$ with its expectation value $\langle\mathbf{S}_i\mathbf{S}_j\rangle = \pm 1/4$ such that $\xi_{ij}^1 = 1, \xi_{ij}^0 = 0$ for ferromagnetic spin order and $\xi_{ij}^1 = \xi_{ij}^0 = 1/2$ for Néel order.

After such a mean-field decoupling corresponding to the ground state in the spin sector, we analyze the ground states of the resulting Hamiltonian for the orbital degrees of freedom, which we approximate as classical vectors. We use an iterative energy minimization combined with simulated annealing techniques (see Methods) to converge the mean-field equations and find the phase diagram summarized in Fig. 4. Panel (a) shows the energy of ferromagnetic and antiferromagnetic spin configurations from which the magnetic phase diagram can be read off. This is given in the upper part of the plot and we find antiferromagnetic ordering with an intermittent ferromagnetic phase at intermediate ratios of $0.1 < J/U < 1/3$. In the lower part of the plot, we show the corresponding subsidiary orbital order. From our simulations, we identify three different configurations of orbital vectors $\boldsymbol{\tau}$, which can be classified according to their projection on a single definite plane in space, shown in the lower left of the plots: (1) ferro-orbital (FO) nematic order[5,6,60–62], where the vectors on all lattice sites align in parallel to the xz-plane. Quantum mechanically, finite values of $\langle\tau_i^{x/z}\rangle$ indicate an imbalance of the occupation of $p_x$ and $p_y$ orbitals, breaking

rotation symmetry and thereby motivating the notion of a nematic state. (2) AFO nematic order; each vector is aligned anti-parallel with its nearest neighbors corresponding to $\langle\tau_i^{x/z}\rangle \neq 0$ on each sublattice, but without finite projections $\tau_i^y$ on individual sites. (3) FO magnetic order; all vectors order along the $y$-axis, such that $\langle\tau_i^y\rangle \neq 0$, which, in the quantum mechanical system, would indicate time-reversal symmetry breaking. The inclusion of quantum fluctuations can change this picture and more exotic ground states may emerge. For example, for our ab initio band structure parameters, a noncollinear spin dimer phase is predicted in a certain range of interaction couplings and even a quantum spin-orbital liquid is found in its proximity[53]. Since these exotic phases primarily occur for weak Hund's coupling and strong orbital anisotropies, the assumptions made for our calculations can therefore be justified for sizable $J_H$ and modest distances to the isotropic $t_\sigma = t_\pi$ point.

## Discussion

We have established that twisted bilayer MoS₂ is a promising platform to realize the orbital anisotropic $p_x$–$p_y$ Hubbard model by employing large-scale ab initio calculations. We find that families of flat bands emerge where the first family of flat bands shows s-orbital character and the second family is an intriguing realization of a strongly asymmetric $p_x$–$p_y$ Hubbard model both on a honeycomb lattice, adding a lattice with nontrivial almost perfectly-flat bands due to destructive interference to the growing list of systems that can be engineered in twisted heterostructures. The symmetry of these flat bands is inherited from the hexagonal lattice formed by the equivalent $B^{Mo/S}$ and $B^{S/Mo}$ regions. At an even smaller angle, the sequence in the family of flat bands found with respect to their orbital character continues. Our analysis shows that the low-energy DFT band structures in this system can

be well captured by a free electron gas model modulated by a simple harmonic potential that has hexagonal ($D_6$) symmetry, which is consistent with a recent study[63]. This simple model further shows that the next family would exhibit a d-orbital character on the honeycomb lattice. Such a lattice would effectively realize a multi-orbital generalization of a Kagome lattice – a prototypical model for quantum spin liquids. However, at such small angles strong relaxation is likely to become dominant, prohibiting access to this regime and potentially spoiling its experimental realization. Currently, the ab initio characterization of such small angles is numerically too exhaustive and this work sparks a direct need for novel computational methods to tackle this question.

Furthermore, our combined exact diagonalization and strong-coupling expansion approaches classify the magnetic and orbital phase diagrams, however, the inclusion of quantum fluctuations stipulates an intriguing avenue of future theoretical research.

Indeed, previous theoretical works provide some evidence for a quantum spin liquid in the SU(4)-symmetric Kugel-Khomskii model on the honeycomb lattice[64], the square lattice as a related system without frustration[65] and studied the role of perturbations that break SU(4) symmetry and isotropy[53] In twisted MoS$_2$, this regime would in fact map to larger twist angles, where the anisotropy of the p$_x$-p$_y$ model is less pronounced, as well as to a regime of vanishing Hund's coupling, placing such a putative quantum spin liquid at the transition between FO nematic and AFO nematic phases.

In addition, by proximity or variations in the chemical composition of the twisted bilayer, it might be possible to induce spin-orbit coupling splitting of the ultra-flat bands at the top and bottom of the asymmetric p$_x$–p$_y$ dispersion. Such a bandgap opening would induce interesting topological properties[66] in a highly tunable materials setting.

## Methods

**Details on ab initio calculations.** We calculate the electronic properties of twisted bilayer MoS$_2$ with ab initio methods based on density functional theory (DFT) as implemented in the Vienna ab initio Simulation Package (VASP)[67]. We employ plane-wave basis sets with an energy cutoff of 550 eV and pseudopotentials as constructed with the projector augmented wave (PAW) method[68]. The exchange-correlation functionals are treated at the generalized gradient approximations (GGA) level[69]. The supercell lattice constants are chosen such that they correspond to 3.161 Å for the $1 \times 1$ primitive cell of MoS$_2$. Vacuum spacing larger than 15 Å is introduced to avoid artificial interaction between the periodic images along the z-direction. Because of the large supercells, a $1 \times 1 \times 1$ k-grid is employed for the ground state and the relaxation calculations. For all the calculations, all the atoms are relaxed until the force on each atom is less than 0.01 eV/Å. Van der Waals corrections are considered with the method of Tkatchenko and Scheffler[70]. We extract the real and the imaginary parts of the DFT wavefunctions with the VASPKIT code[71].

**Details on exact diagonalization.** Exact diagonalization calculations were performed for the electronic tight-binding model in Eq. (1) with Hubbard-Kanamori interactions defined in Eq. (2). All calculations were performed for a two-orbital eight-site cluster with periodic boundary conditions at quarter filling, corresponding to eight spin-1/2 particles in sixteen orbitals. Rotationally symmetric Kanamori interactions are adopted, with $U' = U - 2J$. As the magnitudes of the Hubbard $U$ and Hund's exchange $J$ interactions cannot be reliably predicted for a Moié supercell from first principles, all presented results are shown as a function of $U$, $J$. Calculations of the single-particle Green's functions and local density of states are performed starting from the ground state in the total momentum $K_{tot} = 0$ and total spin $S_z = 0$ sectors, using the Lanczos method and continued-fraction representation, and a spectral broadening (imaginary part of the self-energy) of $\eta = 0.1$ is imposed.

**Details on minimization procedure for classical Hamiltonian.** Metropolis Monte Carlo simulations are a prime tool for the investigation of classical spin models, since they allow for off-diagonal, spatially anisotropic spin couplings to be included, even when one-spin terms, such as magnetic fields, are involved. Here we employ a special variant of the algorithm to the mean-field version of (3), keeping in mind that the 'spins' used in the simulation are approximations to orbital operators $\tau$. First, a lattice site $i$ is randomly chosen, and its respective gradient field

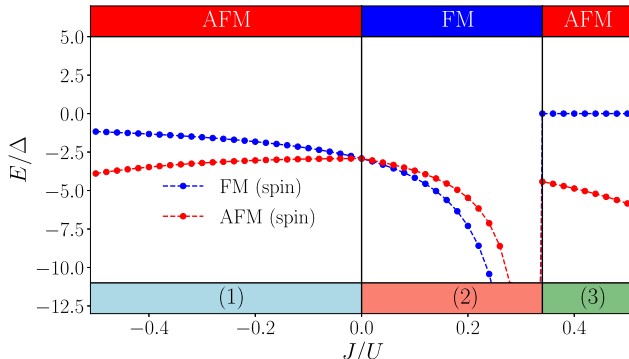

**Fig. 5 Magnetic phase diagram of the strong-coupling Hamiltonian (3) in the isotropic limit $t_\sigma = t_\pi$.** Our results from iterative minimization are in agreement with ref. [52], stabilizing FO nematic order (1) for $J/U < 0$ (see main text) and FO magnetic order (3) (see main text) for $J/U > 1/3$. In the intermediate range of parameters $0 < J/U < 1/3$ the ferromagnetic spin sector is selected, such that, due to vanishing $\xi_{ij}^0$, rotation invariance is restored for the orbital vectors, giving rise to AFM order (2) but without any preferred axis in euclidean space.

$h_i = \nabla_i H$ is computed for the current spin configuration $\{\tau_i\}$. Second, a random orientation $\tau_i'$ for the vector at site $i$ is proposed and the weight

$$g = \min\left(e^{-\beta(\tau_i' - \tau_i)h_i}, 1\right), \qquad (4)$$

is computed for an effective inverse temperature $\beta$. Performing several Metropolis updates with increasing values of $\beta$ we are able to efficiently lower the energy of a random initial configuration, minimizing the odds to converge to a local minimum by only allowing optimal updates (i.e. $\tau_i = -h_i$) right from the start. After $N_a$ sweeps over the full lattice, the so-obtained configuration is ameliorated by $N_o$ optimization sweeps, where the randomly selected spin is rotated anti-parallel to the local gradient field such that the energy is deterministically lowered in every step and we converge as close to the global energy minimum as possible. Hence, this algorithm is reminiscent of Monte Carlo simulations with simulated annealing, but at zero temperature where thermal fluctuations are frozen out.

To benchmark our implementation we have carried out the minimization procedure in the isotropic limit $t_\sigma = t_\pi$ for $N_a = N_o = 10^5$, where the optimization sweeps are terminated when the energy change after one sweep, $\epsilon$, becomes small (usually $\epsilon \leq 10^{-10}$). Mapping out the phase diagram for both the FM, $\langle S_i S_j \rangle = 1/4$, as well as the AFM, $\langle S_i S_j \rangle = -1/4$, spin sector on a lattice with $N = 1250$ spins subject to periodic boundary conditions we find the result in Fig. 5, which is consistent with the one presented in ref. [52]. For $J < 0$ the AFM spin sector has lower energy, with the orbitals forming a ferro-orbital (FO) nematic state where $\langle \tau_i^{x/z} \rangle \neq 0$ and $\langle \tau_i^y \rangle = 0$. For $J > 0$ one finds the FM spin sector (for which the orbital degrees of freedom restore their rotation invariance) to dominate as long as $J < 1/3$, where the AFM sector takes over again and establishes a FO magnetic state, i.e. $\langle \tau_i^{x/z} \rangle = 0$ and $\langle \tau_i^y \rangle \neq 0$.

## Data availability
The raw data sets used for the presented analysis within the current study are available from the corresponding authors on reasonable request.

## Code availability
The tailored developed codes used in this work can be provided from the corresponding author on reasonable request. Ab initio calculations are done with the code VASP (version 5.4.4).

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

## Acknowledgements

We are grateful to Abhay Pasupathy, Cory Dean and Dmitri Basov for fruitful discussions. This work is supported by the European Research Council (ERC-2015-AdG-694097), Grupos Consolidados (IT1249-19), and SFB925. MC and AR are supported by the Flatiron Institute, a division of the Simons Foundation. We acknowledge funding by the Deutsche Forschungsgemeinschaft (DFG, German Research Foundation) under Germany's Excellence Strategy - Cluster of Excellence Matter and Light for Quantum Computing (ML4Q) EXC 2004/1 - 390534769 and Advanced Imaging of Matter (AIM) EXC 2056 - 390715994 and funding by the Deutsche Forschungsgemeinschaft (DFG, German Research Foundation) under RTG 1995 and RTG 2247. Support by the Max Planck Institute - New York City Center for Non-Equilibrium Quantum Phenomena is acknowledged. DK, MMS, and ST acknowledge support from the Deutsche Forschungsgemeinschaft (DFG, German Research Foundation), Projektnummer 277146847 – CRC 1238 (projects C02, C03).

## Author contributions

L.X., D.M.K. and A.R. conceived the project. D.M.K., S.T. and A.R. designed and coordinated the research. L.X., M.C., D.K. and M.S. performed all the simulations. All authors discussed and analyzed the results and contributed to writing the manuscript.

## Funding

## Competing interests

The authors declare no competing interests.
