## [Peer Review File · Nature Communications]

REVIEWER COMMENTS

Reviewer #1 (Remarks to the Author):

This is a review of the manuscript titled "Realization of nearly dispersionless bands with strong orbital anisotropy from destructive interference in twisted bilayer MoS₂" by Xian et al. Overall this is an excellent manuscript which will add a new dimension to the exciting and new physics that can be realized using twisted TMDC structures. The authors have shown that Gamma derived twisted bilayer TMDCs can be doped and will behave as a realization of the anisotropic px-py model. The manuscript is well written and careful and extensive calculations by the authors back their claims sufficiently. I do have a few comments which I would like the authors to address before I would be willing to accept this manuscript.

1. The authors have mentioned that in the presence of spin orbit coupling, the ultra-flat bands can become topologically non-trivial. Could the authors comment on this further? Twisted bilayer MoS₂ is a Gamma point derived twisted homobilayer and as a result spin orbit coupling should not be very important

here. Nevertheless, there should be some effect of the atomic spin orbit coupling. Have the authors done any calculations including these effects? Have they seen these topological non-triviality in their calculations (when including the spin-orbit coupling in their DFT calculation)?

2. The authors have done both exact diagonalization calculations and strong-coupling expansions to study the px-py model. As they have noted in the paper, an extensive study including quantum fluctuation effects is outside the scope of this paper. I agree with them on this aspect. However, I think a few comments about the effects of such quantum fluctuations are definitely in order. For example, can the authors justify the calculations that they have performed without quantum fluctuations?

3. The authors find that the first two valence bands in the twisted homo bilayer structure realize a compressed graphene band structure (near the Dirac point). This has already been pointed out in the paper that the authors have cited -- Ref. 32. While the emphasis of this paper is on the second set of moiré bands, are their bandwidths and band structure in Figure 1 (b) and (c) in agreement with Ref 32?

Reviewer #2 (Remarks to the Author):

The manuscript analyses clearly, in a very detailed and interesting way, the nature of flat bands in twisted bilayer MoS₂. Focus is done on the second groups of lowest energy flat bands, called “ultra-flat” bands. It seems to me that the main original contribution was to show that the ultra-flat bands are well describe by an effective px-py tight-binding model. This makes it possible to better understand the properties of the symmetries of these new states, and to study their magnetic properties using a Hubbard Kanamori Hamiltonian. This work is very interesting and stimulating in the very active field of Moiré flat bands. It should give rise to new experiments to verify these predictions. That is why I think this manuscript deserves to be published in a very good journal. However, I have a three remarks/questions to which the authors should be able to respond without difficulty.

(1) The author repeatedly stresses that the origin of the "ultra-flat" bands is different from the usual flat bands in twisted bilayers (such as twisted bilayer graphene or lower energy flat bands in MoS₂ twisted bilayers). And so, in the abstract, they wrote: "The origin of these dispersionless bands, is similar to that of the flat bands in the prototypical Lieb or Kagome lattices and co-exists with the general band flattening at small twist angle due to the Moire interference." This sentence suggests that the ultra-flat bands are not due to Moiré structure, whereas it seems to me that both type of flat bands come from Moiré pattern, and that the difference between flat and ultra-flat bands lies in their symmetry properties. So I do not understand why the author insists that the natures of these two types of Moiré flat bands are so different.

(2) Several times the authors write that the states at the top of the valence band are p S character and dz² Mo character (see for instance pages 7 and 9). I agree that the p S orbitals have an essential role in the opening of the gap in TMDs, but p S orbitals acts as a perturbation of the d Mo states, and the states around the gap are mainly d Mo states. This is not essential for the purpose of this article, but it seems to me that this needs to be clarified.

(3) This work shows, very clearly, that an effective px-py tight-binding model describes the ultra-flat bands very well. This model include the nearest neighbor hopping terms and the next nearest neighbor hopping terms. To study correlation and magnetic properties, the next nearest neighbor hopping terms are neglected. It would be more convincing to justify this approximation by showing, for example, a comparison of the (non-interacting) bands with and without the next nearest neighbor hopping terms.

Reviewer #3 (Remarks to the Author):

The reviewed manuscript presents a theoretical and computational study of flatband physics in twisted bilayer MoS₂. The focus is on the second set of moire valence bands, which can be captured by an effective p-orbital tight-binding model on a honeycomb lattice. Effects of many-body interaction are also studied within the effective lattice model. I have the following questions.

(1) There is no symmetry analysis that shows clearly that the second set of moire valence bands can be mapped to p-orbital tight-binding model. Such an analysis is given in arXiv:2008.01735.

(2) To access the second set of valence bands, the first set of valence bands needs to be emptied first, which can lead to many-body renormalization of the moire potential and moire band structure. In other words, the moire bands can be filling factor dependent. It is unclear whether the second set of valence bands can still be described by the p-orbital tight-binding model after the first set of valence bands is emptied.

(3) The Hamiltonian with local interactions in Eq. (2) is not justified. First, the long-range nature of Coulomb potential has been shown to be important given the observation of generalized Wigner crystal in twisted bilayer TMDs. Second, the Wannier states are not constructed explicitly. Therefore it is unclear whether Eq. (2) captures all important local interactions.

(4) "quarter filling" needs to be explicitly defined. I failed to find an explicit definition of filling factor.

Given similar studies in the literature, for examples, PRL 121, 266401 (2018) and Nano Lett. 19, 4934 (2019), I am not fully convinced that this manuscript should be published in a highly selective journal like Nature Communications.

Response to Referee #1, (1)

The authors have mentioned that in the presence of spin orbit coupling, the ultra-flat bands can become topologically non-trivial. Could the authors comment on this further? Twisted bilayer MoS2 is a Gamma point derived twisted homobilayer and as a result spin orbit coupling should not be very important here. Nevertheless, there should be some effect of the atomic spin orbit coupling. Have the authors done any calculations including these effects? Have they seen these topological non-triviality in their calculations (when including the spin-orbit coupling in their DFT calculation)?

We thank the referee for making this excellent point. The Γ -point states that are responsible for the flat bands are not affected by spin-orbit coupling (SOC). We have now included a new figure in the supplemental information (Fig. S3) showing the band structures calculated with SOC at 3.15 degrees. The calculated band structures for the top valence bands up to 0.4 eV below the band edges are identical for calculations with and without SOC. Therefore, in this materials, as the referee correctly pointed out, we cannot achieve topological non-triviality through SOC in the pristine system. Nevertheless, we expect that substrate engineering can be used to introduce SOC splitting into these bands and turn the ultraflat band states topologically non-trivial. We added this discussion to the main text.

Response to Referee #1, (2)

The authors have done both exact diagonalization calculations and strong-coupling expansions to study the px-py model. As they have noted in the paper, an extensive study including quantum fluctuation effects is outside the scope of this paper. I agree with them on this aspect. However, I think a few comments about the effects of such quantum fluctuations are definitely in order. For example, can the authors justify the calculations that they have performed without quantum fluctuations?

We agree with the referee that one should definitely comment about possible effects of quantum fluctuations and we appreciate the opportunity to clarify on that point. In the manuscript we cited recent research (Ref. 49), which demonstrates that besides the AFO nematic state we identified, fluctuations may stabilize a non-collinear spin dimer phase as well as a spin orbital liquid. However, these exotic phases seem to be most prominent for strong orbital anisotropy, i.e. a large difference between the t_σ and t_π hoppings, favoring the dimer phase, and weak Hund's coupling J_H , where the suggested model bares resemblance with a SU(4) symmetric Kugel-Khomskii Hamiltonian. To justify the approximations made in our work, we have therefore added the sentence "*Since these exotic phases primarily occur for weak Hund's coupling and strong orbital anisotropies, the assumptions made for our calculations can therefore be justified for sizable J_H and modest distances to the isotropic $t_\sigma = t_\pi$ point.*" to our manuscript.

Response to Referee #1, (3)

The authors find that the first two valence bands in the twisted homo bilayer

structure realize a compressed graphene band structure (near the Dirac point). This has already been pointed out in the paper that the authors have cited – Ref. 32. While the emphasis of this paper is on the second set of moiré bands, are their bandwidths and band structure in Figure 1 (b) and (c) in agreement with Ref 32?

Note that when Ref. 32 was first posted on arXiv (arXiv:1908.10399v1) , the authors did not point out the compressed graphene feature. Only a couple months later after we posted our work on arXiv (arXiv:2004.02964), they updated a second version (which is similar to the one eventually published in PRB) adding such discussion. These findings were thus obtained independently. Our results and those in Ref. 32 on the second set of moire bands share similar features. The remaining quantitative differences are most likely due to different structural relaxation scheme adopted in the two works: we relaxed all the atoms with DFT calculations while they used a more approximate force field approach for the relaxation. We therefore believe our full DFT calculations delivers a more accurate picture of the flat band features of the system.

Response to Referee #2, (1)

The author repeatedly stresses that the origin of the "ultra-flat" bands is different from the usual flat bands in twisted bilayers (such as twisted bilayer graphene or lower energy flat bands in MoS2 twisted bilayers). And so, in the abstract, they wrote: "The origin of these dispersionless bands, is similar to that of the flat bands in the prototypical Lieb or Kagome lattices and co-exists with the general band flattening at small twist angle due to the Moire interference." This sentence suggests that the ultra-flat bands are not due to Moiré structure, whereas it seems to me that both type of flat bands come from Moiré pattern, and that the difference between flat and ultra-flat bands lies in their symmetry properties. So I do not understand why the author insists that the natures of these two types of Moiré flat bands are so different.

We thank the referee for giving us the opportunity to clarify this. There are two levels of discussion in moire 2D materials. The first level is the discussion of moire flat bands. In moire materials, the interlayer interaction and/or local reconstruction are modulated by the large scale moire pattern, leading to the formation of large scale moire potentials. These moire potentials lead to the formation of moire flat bands. In moire semiconductors, the band width of moire flat bands at the band edges usually decreases monotonically with twist angles (or inversely with moire supercell size). These moire flat bands can usually be described by some effective theoretical models. The second level of discussion is about which effective models can be realized within moire flat bands. One typical example is the triangular Hubbard model that is realized in moire TMD heterostructures. The band width of these effective models usually also depends on the twist angles, which determines the effective hopping amplitude. However, there is one type of special effective models that have flat bands simply due to the geometry effect of the models. These are, e.g., the Lieb and the Kagome lattice models, and also the px-py honeycomb model we discussed in this work.

In these models, there is at least one ultra-flat band due to destructive interference of the different paths electrons can move on in the lattice and the band width of these ultra-flat bands is independent of the hopping amplitude of the whole model. Moreover, such ultra-flat band(s) have non-trivial band topology when there are spin-orbit couplings.

Therefore, we want to make a point here that in our work, we show that on the first level, twisted bilayer MoS₂ also hosts some moire flat bands at the top of the valence bands; and most importantly on the second level, one set (the second set) of the moire flat bands realize the special px-py honeycomb model, which host ultra-flat bands on its own. This special px-py honeycomb model has not been realized (or recognized) before in other moire materials and it represented a new type of interesting effective correlated model that can be studied in moire 2D materials. As we pointed out in our manuscript, due to the multi-orbital characters and high level of degeneracy, this new type of effective model has many interesting properties to be explored.

Response to Referee #2, (2)

Several times the authors write that the states at the top of the valence band are p S character and dz² Mo character (see for instance pages 7 and 9). I agree that the p S orbitals have an essential role in the opening of the gap in TMDs, but p S orbitals acts as a perturbation of the d Mo states, and the states around the gap are mainly d Mo states. This is not essential for the purpose of this article, but it seems to me that this needs to be clarified.

We thank the referee for pointing this out. Her/his comments have clearly improved our presentations and we have now included a new figure (Fig. S2) in the Supplementary Information showing the atomic characters of bilayer MoS₂ (see below). The referee's comment is entirely correct for monolayer TMDs where the top of the valence bands is located at the K points. The states at the top of the valence bands at the K points are indeed dominated by the Mo d states. But for bilayer MoS₂ (and some other TMDs like WS₂ and MoSe₂), the top of the valence bands is located at the Γ points. These states at the top of the valence bands are contributed to by both S p orbitals and Mo d orbitals as can now be seen in Fig. S2. Moreover, the interlayer interaction in a multilayer (bilayer) system are dominated by the coupling between S p_z orbitals in adjacent layers (e.g., see discussions in section V of Phys. Rev. B 92, 205108 (2015)). Therefore, although it is possible to construct a simple tight-binding model for monolayer TMDs with only Mo d orbitals [Phys. Rev. B 88, 085433 (2013)], it is crucial to include the X(X=S,Se,Te) p orbitals in the discussion of the electronic properties of bilayer TMDs [Phys. Rev. B 92, 205108 (2015)].

Response to Referee #2, (3)

This work shows, very clearly, that an effective px-py tight-binding model describes the ultra-flat bands very well. This model include the nearest neighbor hopping terms and the next nearest neighbor hopping terms. To study correlation and magnetic properties, the next nearest neighbor hopping terms are neglected. It would be more convincing to justify this approxima-

tion by showing, for example, a comparison of the (non-interacting) bands with and without the next nearest neighbor hopping terms.

We thank the referee for pointing this out. As stated in the main text, in the px-py model which are fitted to the DFT flat bands, the parameters for the next nearest neighbour hopping are about one order of magnitude smaller than those of the nearest neighbour hopping ($t_\pi = 0.25t_\sigma$, $t_\sigma^N = 0.07t_\sigma$ and $t_\pi^N = -0.04t_\sigma$). Therefore, when discussing the correlation and magnetic properties in the strong coupling regime (i.e., $U > t$), we neglected the next nearest neighbour hopping terms. As suggested by the referee, we include a new figure (Fig. S3) in the supplementary information to check on the severity of this approximation.

Response to Referee #3, (1)

There is no symmetry analysis that shows clearly that the second set of moire valence bands can be mapped to p-orbital tight-binding model. Such an analysis is given in arXiv:2008.01735.

We thank the referee for pointing this out. Indeed, our work characterized the character of the flat bands in twisted bilayer MoS₂ mainly from the band structure and wavefunction/charge density analysis and a more detailed symmetry analysis is done in a later work in arXiv:2008.01735. We have added discussion and included the citation suggested by the referee to accommodate this.

Response to Referee #3, (2)

To access the second set of valence bands, the first set of valence bands needs to be emptied first, which can lead to many-body renormalization of the moire potential and moire band structure. In other words, the moire bands can be filling factor dependent. It is unclear whether the second set of valence bands can still be described by the p-orbital tight-binding model after the first set of valence bands is emptied.

We thank the referee for pointing this out. We agree with the referee that one needs to empty the first set of valence bands to access the second set of valence bands. This will introduce additional renormalization of the band structure. To evaluate such effect, we performed additional first principles calculations of twisted bilayer MoS₂ at 2.28 degrees with 4-electron hole doping, which empties the first set of valence bands. The resulting band structure is now shown in Fig. S5 and aligned with original one without doping for comparison. With doping, although the lower branch of the px-py bands is indeed modified quantitatively, the overall shape remains. Doping doesn't simply correspond to a trivial fermi level shift as done in many other works, but in this case the nature and symmetry of the second set of bands is kept. The correction is much smaller than the total bandwidth. In the strong-coupling regime we discussed in the main text, such changes will modify the effective parameters only slightly. Moreover, as the doping density required to empty the first set of bands decreases with twist angle, we expect such effect will be even smaller in smaller twist angles. We have now added a discussion of this important point.

Response to Referee #3, (3)

The Hamiltonian with local interactions in Eq. (2) is not justified. First, the long-range nature of Coulomb potential has been shown to be important given the observation of generalized Wigner crystal in twisted bilayer TMDs. Second, the Wannier states are not constructed explicitly. Therefore it is unclear whether Eq. (2) captures all important local interactions.

We concur with the referee that longer-ranged interactions neglected in our analysis can play an important role at certain fillings, and the filling-dependent phase diagram is likely very rich. However, for our choice of commensurate quarter filling, any longer-ranged component of the Coulomb interaction at strong coupling will serve merely to slightly renormalize the effective spin-orbital interactions of the resulting Kugel-Khomskii model. Moreover, the leading order contribution will come from longer-ranged density interactions, which renormalize all virtual exchange processes equally, hence leave the phase diagram unchanged. We therefore neglect these in our analysis. The *local* interactions [Eq. (2)] used in our analysis are of the standard Kanamori type, constrained by rotational symmetry. We have added a clarifying sentence to the main text.

Response to Referee #3, (4)

"quarter filling" needs to be explicitly defined. I failed to find an explicit definition of filling factor.

We thank the referee for pointing out this omission, and have added an explicit definition to the main text. Quarter filling entails one electron per Moiré unit cell.

Response to Referee #3, (5)

Given similar studies in the literature, for examples, PRL 121, 266401 (2018) and Nano Lett. 19, 4934 (2019), I am not fully convinced that this manuscript should be publised in a highly selective journal like Nature Communications.

We thank the referee for his/her critical assessment of our work. We would like to stress that our findings are far beyond the prior literature. It extends the class of models that can be realized effectively by moire engineering by adding the asymmetric px-py honeycomb lattice model to the list. The interacting ground states and physical properties that can be explored in this model is distinctly different from what has been studied in the prior moiré systems. In particular, this model shows interesting interference effects similar to that of the prototypical Lieb or Kagome lattice and we establish that twisted bilayer MoS₂ could be the first controllable condensed-matter based inroad into this highly intriguing physics. As such we would like to maintain the fact that our work is significantly pushing the frontier of the highly volatile field of moire materials.

REVIEWER COMMENTS

Reviewer #1 (Remarks to the Author):

The authors have addressed all my concerns adequately.

I would like to clarify that, with respect to point #3, I did not imply that the authors here did not arrive at the compressed graphene feature independently. Clearly, in the paper Ref 32, the authors did not realize the other features of the band structure (pxpy) and sd2 etc. A comparison would be ideal but if the authors feel that this takes away any credit from their independent discovery, not discussing is also fine by me. As one of the author of Ref 32, I would like to assure the authors that the compressed graphene finding in that paper were arrived at independently.

I recommend the acceptance of the manuscript in the present form.

Reviewer #2 (Remarks to the Author):

I find that the authors answer my remarks/questions in a satisfactory manner, in particular with new figures S2 and S4 in the Supplementary Information. It seems to me that they also answer properly the questions of the other referees. So I think this manuscript deserves to be published in Nature Communication.

Reviewer #3 (Remarks to the Author):

The authors have carefully addressed referees' questions. Before I recommend publication of the manuscript at Nature Communications, I still have 3 additional comments.

(1) In Fig. S5, the band structure with and without doping is calculated. It is surprising to find that the gap between the first and the second moire valence bands is reduced after doping. I expect that this

gap becomes enhanced after doping because of Hartree and exchange self energy. Could the authors comment on why the gap is reduced?

(2) I am still confused by the filling factor. In Fig. 4b, it looks to me that there is one carrier per sublattice of the honeycomb lattice, i.e., there are two (not one) carriers per moire unit cell. The authors may need to double check the filling factor.

(3) The following references might be of interest to the authors: Phys. Rev. Lett. 121, 026402 (2018), Phys. Rev. Lett. 122, 086402 (2019).

Point-by-Point Response

Reviewer #1 Comments:

The authors have addressed all my concerns adequately.

I would like to clarify that, with respect to point #3, I did not imply that the authors here did not arrive at the compressed graphene feature independently. Clearly, in the paper Ref 32, the authors did not realize the other features of the band structure ($p_x p_y$) and sd_2 etc. A comparison would be ideal but if the authors feel that this takes away any credit from their independent discovery, not discussing is also fine by me. As one of the author of Ref 32, I would like to assure the authors that the compressed graphene finding in that paper were arrived at independently.

I recommend the acceptance of the manuscript in the present form.

We thank the referee for their positive evaluation. Our intention was not to leave out such a comparison, but apparently have failed to present it in a satisfactory manner. We have included additional information to further improve the manuscript.

Reviewer #2 Comments:

I find that the authors answer my remarks/questions in a satisfactory manner, in particular with new figures S2 and S4 in the Supplementary Information. It seems to me that they also answer properly the questions of the other referees. So I think this manuscript deserves to be published in Nature Communication.

We are grateful to the referee for their positive evaluation and would like to thank them again for the helpful feedback which has clearly improved our work.

Reviewer #3

The authors have carefully addressed referees' questions. Before I recommend publication of the manuscript at Nature Communications, I still have 3 additional comments.

We thank the referee for again taking the time to critically read our work. We have addressed the additional remarks of them in the manuscript.

(1) In Fig. S5, the band structure with and without doping is calculated. It is surprising to find that the gap between the first and the second moiré valence bands is reduced after doping. I expect that this gap becomes enhanced after doping because of Hartree and exchange self energy. Could the authors comment on why the gap is reduced?

From the simple model of Ref. 63 it follows that if the depth of the moiré potential is reduced, the band width will increased and band gaps between different sets of flat bands will decrease. Our result is qualitatively consistent with such picture, indicating that the moiré potential is effectively reduced by the doping. Therefore, we think that the doping introduces additional electrostatic potential centered at the $B^{S/Mo}$ and the $B^{Mo/S}$ regions where the charge density of the states of the first

flat band is localized. This slightly compensates the interlayer moiré potential and effectively reduces it.

(2) I am still confused by the filling factor. In Fig. 4b, it looks to me that there is one carrier per sublattice of the honeycomb lattice, i.e., there are two (not one) carriers per moire unit cell. The authors may need to double check the filling factor.

We thank the referee for catching this, and pointing out that our statement on filling is confusing, and have amended the relevant paragraph accordingly. Indeed, as the unit cell hosts two sublattices with two orbitals each for spinful electrons, quarter filling entails one electron per sublattice, meaning two electrons in the honeycomb unit cell.

(3) The following references might be of interest to the authors: Phys. Rev. Lett. 121, 026402 (2018), Phys. Rev. Lett. 122, 086402 (2019).

We thank the referee for pointing out these interesting and very relevant references, which we have included in our manuscript now.